# Nanoimprinted Plasmonic Crystals for Cost-Effective SERS Identification of Methylated DNAs

**DOI:** 10.3390/s24144599

**Published:** 2024-07-16

**Authors:** Daiki Kawasaki, Ryosuke Nishitsuji, Tatsuro Endo

**Affiliations:** 1Metamaterials Laboratory, RIKEN Cluster for Pioneering Research, 2-1 Hirosawa, Wako, Saitama 351-0198, Japan; ipytbwas@icloud.com; 2Department of Information Networking, Graduate School of Information Science and Technology, Osaka University, 2-8 Yamadaoka, Suita, Osaka 565-0871, Japan; nishitsuji@ist.osaka-u.ac.jp; 3Department of Applied Chemistry, Graduate School of Engineering, Osaka Metropolitan University, Sakai, Osaka 599-8531, Japan

**Keywords:** DNA methylation, SERS, plasmonic crystals, nanoimprint lithography, *APC* gene

## Abstract

The development of a cost-effective and rapid assay technique for the identification of DNA methylation is one of the most crucial issues in the field of biomedical diagnosis because DNA methylation plays key roles in human health. The plasmonic crystal-based surface-enhanced Raman spectroscopy (SERS) technique is promising for the realization of such an assay method owing to its capability of generating uniformly enhanced electric fields to achieve high reproducibility and accuracy in SERS assays. However, the time and technical costs of fabricating plasmonic crystals are high, owing to the need for nanofabrication equipment. In this study, we developed nanoimprinted plasmonic crystals for cost-effective and rapid DNA methylation assays. Our plasmonic crystals identified methylated DNA with the 40-base pair adenomatous polyposis coli (APC) gene sequence, which is correlated with cell growth and cancer cells.

## 1. Introduction

The development of a cost-effective technique for the rapid detection and identification of methylated DNA is crucial in biomedical diagnosis because highly methylated DNA is considered to be related to the development and progression of inherited genetic diseases and cancers [1,2,3]. The conventional gold standard method for DNA methylation assays is the bisulfite conversion assay. The bisulfite assay can quantify methylation in DNA but is costly in terms of time and technical requirements and has the intrinsic problem of giving false-positive results [4,5,6]. Other techniques for affinity assays with methylated DNA-recognizable proteins and enzyme-based assays have been developed; however, these techniques do not simultaneously satisfy the requirements of speed, high accuracy, and sensitivity [7,8]. As an alternative technique to meet these demands, plasmonic nanostructure-based surface-enhanced Raman spectroscopy (SERS) has been the focus of research because the SERS spectrum of methylated DNA includes information on the methylation quantity and state, which can be obtained from label-free DNA samples within a few minutes or even seconds [9,10,11]. Therefore, SERS techniques are promising assay methods for methylated DNA detection. However, SERS techniques have not been widely used in practical medical diagnosis mainly for two reasons: first, low accuracy and reproducibility due to non-uniform SERS activating hot spots on the plasmonic substrate, which is usually fabricated by bottom-up processes such as gold nanoparticle self-assembly; and second, the high fabrication costs of nanostructure-arrayed plasmonic substrates for high accuracy, which are usually fabricated by a top-down process with nanofabrication equipment [12,13,14,15]. These two factors have always been in a tradeoff relationship, and it remains challenging to develop cost-effective and highly accurate SERS techniques for DNA methylation assays. Although the designed plasmonic arrayed system is expensive to fabricate, it is a crucial tool for SERS assays because it offers a designed and uniformly enhanced electric field for intense Raman scattering [11,13]. Thus, the development of a cost-effective plasmonic arrayed substrate is expected to open the door to rapid and accurate methylated DNA assay techniques.

Nanoimprinting techniques can be used to solve this problem. Using the nanoimprint technique, a uniformly arrayed plasmonic nanostructure surface can be cost-effectively fabricated with only one original mold [16,17,18,19,20]. Several recent studies demonstrated that nanoimprinted plasmonic substrates work as SERS platforms [21,22]. In this study, complementary nanoimprinted arrayed plasmonic surfaces composed of nanodisks under or on nanoholes were developed for the rapid and accurate identification of methylated DNA with the adenomatous polyposis coli (APC) gene sequence, which is generally related to the regulation of important events in cell growth and cancer cells [23]. The former are called plasmonic nanoholes (PNHs), and the latter are called plasmonic nanodisks (PNDs). First, the PNHs and PNDs were fabricated and characterized, followed by optical characterization. To apply the PNHs and PNDs to SERS with 785 nm laser excitation, the plasmonic resonance wavelengths of the PNHs and PNDs were adjusted to 785 nm by tuning the thickness of the gold layer. After the characterization and optimization of the PNHs and PNDs, the SERS properties were evaluated using 4-mercapto benzoic acids (4-MBA). Finally, the capability of methylated DNA identification was investigated using the resonant wavelength-optimized PNHs and PNDs.

## 2. Materials and Methods

### 2.1. The Fabrication of the Plasmonic Crystals

A cyclo-olefin polymer (COP)-based hole array-structured film, which we developed in a previous study, was used as the mold for nanoimprint lithography [20]. First, the COP mold was cleaned with 2-propanol (Kanto Chemical Co. Inc., Tokyo, Japan) and ultrapure water and then dried at room temperature (20–25 °C). The Au layer (thickness t = 30, 40, 50, 70, and 100 nm) was thermally deposited onto the COP mold. This Au hole array nanostructure was used as PNHs. The PNDs were prepared by attaching the deposited Au layer to a glass substrate using a UV-curable polymer (NOA81, Norland Products Inc., Cranbury, NJ, USA), followed by the dissolution of the COP mold in limonene (Kanto Chemical Co. Inc., Tokyo, Japan). The nanostructures were observed using field-emission scanning electron microscopy (FE-SEM) (SU8010, Hitachi, Ibaraki, Japan) at an acceleration voltage of 10 keV.

### 2.2. Optical Characterization

The experimental optical characterization of the plasmonic crystals was performed by measuring the refraction spectra using our purpose-built spectroscopic setup (Appendix A). The optical properties of the plasmonic crystals were characterized by numerical calculations using the FDTD method. A plane wave was used as the light source, electric field profiles and reflection intensity were calculated by field monitors, periodic boundary conditions were set in the x-y direction, and a perfect matching layer was set in the z direction.

### 2.3. Raman Spectroscopy

In the evaluation of the SERS properties of PNHs and PNDs in Section 3.2, after the PNHs and PNDs were washed with ethanol, each substrate was immersed in a 1 M ethanolic solution (2 mL) of 4-mercaptobenzoic acid (4-MBA) (>97%, Tokyo Chemical Industry Co., Ltd., Tokyo, Japan) for one day to modify 4-MBA onto the gold nanostructure surface. The modified substrate was washed with ethanol to remove unbound 4-MBA and dried using an air blower. All SERS spectra were measured using a laser confocal Raman microscope (RAMAN-11, Nanophoton, Osaka, Japan) with a 785 nm laser. SERS measurements were performed under the following conditions: 50× objective lens (N.A. = 0.8), 1 mW laser power, and 60 s integration time. In the evaluation of methylated DNA identification capability, after the PNHs and PNDs were washed with ultrapure water, each substrate was immersed in 1 μM phosphate-buffered saline (PBS) solution (pH: 7.4) (100 μL) of 5′ thiolated probe DNAs for one hour to modify probe DNAs onto the gold nanostructure surface. After washing the substrate with PBS and ultrapure water, each substrate was immersed in a PBS solution of target DNAs (1 μM) for one hour to facilitate the hybridization of target DNAs with probe DNAs. The DNA sequences used in this study are listed in Appendix A (all purchased from BEX Co., Ltd. (Toyota, Japan)).

## 3. Results and Discussion

### 3.1. Fabrication and Optical Characterization of Plasmonic Crystals

Complementary plasmonic crystals, PNHs, and PNDs were fabricated from arrayed nanohole-patterned polymer-based films using nanoimprinting techniques (Figure 1). The geometries of the nanostructured surfaces of the PNHs and PNDs with a 100 nm thick gold layer are shown in Figure 1. The diameter of each hole or disk was approximately 110 nm, the pitch was approximately 440 nm, and the depth or height of each hole or disk was approximately 190 nm. Our plasmonic crystals could be fabricated from massively producible and reusable polymer-based molds. Once the original mold was prepared, the fabrication of plasmonic crystals could be performed without expensive and time-consuming equipment, which enables high-throughput and cost-effective fundamental experiments and practical applications in biomedical fields. After preparing PNHs and PNDs with gold layers of various thicknesses, their optical properties were characterized by experimentally measuring the microscopic reflection spectrum and numerically calculating the reflection spectrum and electric field distribution on the x-z cross-section of the nanostructure (Figure 2 and Appendix A). The experimental reflection spectra of the PNHs and PNDs with gold layers of varying thicknesses are displayed in Figure 2a,b. Two dips were observed in the reflection spectra of both the PNHs and PNDs. The plasmonic modes of the two dips are indicated by the symbols i and ii, respectively. Mode i is the surface lattice resonance (SLR) mode excited on holes or disks supported by under-disks or holes, for which the resonant wavelength can be tuned by the thickness of the gold layer. On the other hand, mode ii is based on the plasma frequency and is difficult to tune. To apply plasmonic crystals to SERS with 785 nm laser excitation, mode i was tuned to 785 nm by varying the thickness of the gold layer in the range of 30–100 nm. The dashed line in the spectra marks 785 nm and indicates an optimal gold layer thickness of 40 nm with PNHs and 30 nm with PNDs. The electric field enhancements by the PNHs and PNDs were maximized at the optimal thicknesses (Appendix A). In addition, PNHs barely enhance the electric field when mode i blue-shifts from 785 nm with an increase in the gold layer thickness, whereas PNDs can enhance the electric field even when mode i moves away from 785 nm, which can be seen by comparing the electric field distributions of PNHs and PNDs with a 100 nm thick gold layer. This is because the incident wavenumber along the M-direction in the reciprocal lattice, where the pitch is approximately 762 nm, can be diffracted onto the PNDs’ surface on the holes. Therefore, photonic rather than plasmonic enhancement in the near-field was generated by the surface of the PNDs. In the next section, we evaluate the SERS properties of PNDs and PNHs using a common molecule with an aromatic ring, 4-MBA.

### 3.2. Evaluation of SERS Properties of Plasmonic Crystals

In this section, the enhancement in the Raman scattering of 4-MBA by the PNHs and PNDs is evaluated. Generally, the Raman scattering of molecules with aromatic rings is relatively high because of lower intramolecular thermal deactivation, the Raman property of which enables an accurate evaluation of the Raman enhancement factor by the plasmonic surface. PNHs and PNDs with a 10 nm thick gold layer, which barely enhanced the electric field but enabled the immobilization of 4-MBA on the surface via thiolate bonding, were prepared for the evaluation of Raman enhancement by arrayed nanohole or nanodisk patterns. To evaluate the correlation between the enhanced Raman intensity and electric field enhancement, which is related to the thickness of the gold layer, the typical Raman peak intensity around 1080 cm^−1^ of the PNHs and PNDs with gold layers of various thicknesses was compared (Figure 3). The enhanced Raman intensity is reflected by the electric field enhancement, as shown in Figure 2 and Appendix A. The Raman intensity reached its maximum when the gold thickness was at its optimum value, as shown in Appendix A. As previously discussed in this section, even PNDs with a 10 nm thick gold layer that barely generates plasmonic enhancement could enhance the Raman intensity, which could be attributed to the photonic (diffractive) surface mode. Therefore, the enhancement in Raman scattering by the PNDs was partially attributed to the photonic mode, regardless of the gold layer thickness, whereas that by the PNHs was completely attributed to plasmonic enhancement. In the following section, the capability of PNHs and PNDs with optimized gold layer thicknesses (PNHs: 40 nm and PNDs: 30 nm) to identify DNA methylation by SERS is discussed.

### 3.3. Identification of DNA Methylations by SERS

In this section, the capability of PNHs and PNDs to identify DNA methylation with *APC* sequences was evaluated. According to the results in Section 3.2, the PNDs had a higher Raman scattering enhancement capability than the PNHs when their gold layers were optimized to 40 and 30 nm for the PNHs and PNDs, respectively. Three methylated cytosines in 40 base pairs of *APC* sequences were used in this experiment (Appendix A). Considering the practical application of plasmonic crystals to a commonly used protocol in medical diagnosis, the target single-strand DNA (1 μM) was hybridized with unmethylated probe DNA immobilized on the plasmonic crystal surface; therefore, only one strand had three methylations in double-stranded DNA. First, the enhanced Raman spectra of the methylated and unmethylated *APC* sequences (1 μM) were measured in 20 random locations with each of the four samples of PNHs and PNDs (Figure 4a–d). The spectra in Figure 4a–d were corrected for the baselines using the software (RAMAN Imager version 2; Nanophoton, Osaka, Japan). In the case of methylated DNAs, a Raman peak around 730 cm^−1^, which is reported as one of the typical Raman peaks in methylated DNAs [10], was clearly observed for PNHs, whereas it was barely observed for PNDs. On the other hand, another typical Raman peak around 1330 cm^−1^ was difficult to observe. In the case of unmethylated DNAs, the typical Raman peak at approximately 730 cm^−1^ was not observed for either PNHs or PNDs. Therefore, focusing on the Raman peak at 730 cm^−1^ with PNHs, the methylation of DNA could be identified. According to a previous report [24], it was considered that the Raman peak at 730 cm^−1^ and 1330 cm^−1^ can be attributed to the ring breathing vibration of adenine and guanine, respectively. The authors mentioned that the ring breathing vibration of adenine at 730 cm^−1^ is affected by the methylation of cytosine and showed that the Raman peak intensity at 730 cm^−1^ of cytosine-methylated DNA was higher than that of unmethylated DNA. Since the raw Raman spectrum was too noisy to clearly identify the Raman peaks attributed to methylation, the modified Raman spectrum was obtained by subtracting the Raman spectrum of bare PNHs without DNAs from that of PNHs with DNAs after smoothing the spectra using the Savitzky−Golay algorithm with a second-degree polynomial and window size of 5 (Figure 4e,f) [25]. Smoothing using the Savitzky−Golay algorithm not only reduced noise in the spectra but also clarified peaks. In the subtract process, removing the signals from bare PNHs clarified the peaks derived from the DNA methylation. The subtracted Raman spectrum of methylated DNAs clearly displayed two Raman peaks at approximately 730 cm^−1^ and 1330 cm^−1^ which were not observed in the subtracted Raman spectrum of unmethylated DNAs. Contrary to our expectation from the Raman enhancement evaluation described in Section 3.2, PNHs had a higher capability for the identification of DNA methylation than PNDs, according to the Raman peaks at 730 cm^−1^. With PNDs, the intensity of the broad Raman peaks around 1330 cm^−1^, which is commonly attributed to DNA molecules, was comparable or even slightly higher than that with PNHs, in agreement with the results of the Raman enhancement properties shown in Figure 3c. However, narrow methylation-related Raman peaks were not observed for PNDs. This result can be attributed to the difference in the attribution of “plasmonic” and “photonic” enhancement between PNHs and PNDs. As mentioned above, PNHs enhanced the Raman scattering of 4-MBA in the pure plasmonic mode, while PNDs were enhanced by plasmonic and photonic modes. In the case of highly efficient Raman scattering modes, such as S-C bonding beside the aromatic ring, the intensity increases with the electric field enhancement factor, regardless of the plasmonic or photonic mode. On the other hand, in the case of a low-efficiency Raman scattering mode, such as bonding in DNA molecules, it is considered that the intensity is affected not only by electric field enhancement but also by the thermal diffusion of plasmonically activated electrons in the molecules and gold surface. Thermal diffusion in open plasmonic systems such as PNHs is expected to be much higher than that in closed plasmonic systems such as PNDs. These results suggest that open plasmonic systems such as nanoholes work better for the SERS of biomolecules, even though the electric field enhancement is lower than that of closed plasmonic systems such as nanodisks.

## 4. Conclusions

In this study, we demonstrated the rapid detection of methylated DNA in *APC* sequences using nanoimprinted plasmonic crystals. The plasmonic resonance wavelengths of the PNHs and PNDs were adjusted to 785 nm by tuning the thicknesses of the Au layers. The evaluation of the SERS properties of PNHs and PNDs revealed that PNHs could enhance molecular Raman scattering by their plasmonically enhanced electric field, whereas PNDs could enhance it by their plasmonic and photonic resonance modes. In the evaluation of the capability of methylated DNA identification, PNHs could detect the methylation of the *APC* sequence, whereas PNDs could not. Nevertheless, the 4-MBA Raman enhancement property of PNHs was lower than that of PNDs. This is attributed to the difference in the enhancement effect between the PNHs and PNDs, as well as the thermal diffusion efficiency. In this study, the resonant wavelength was tuned by changing the thickness of the gold layer. To maximize the SERS properties of our imprinted plasmonic crystals, the diameter and pitch should be optimized to increase the plasmonic resonance quality by designing a master mold. Our imprinted plasmonic crystals are promising for cost-effective SERS-based DNA methylation assays for practical diagnosis.

## Figures and Tables

**Figure 1 sensors-24-04599-f001:**
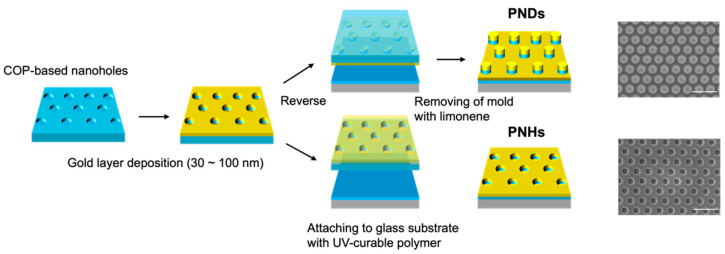
Schematics of nanoimprint-based fabrication process, and SEM image of PNHs and PNDs. Scale bar in SEM image represents 1 μm.

**Figure 2 sensors-24-04599-f002:**
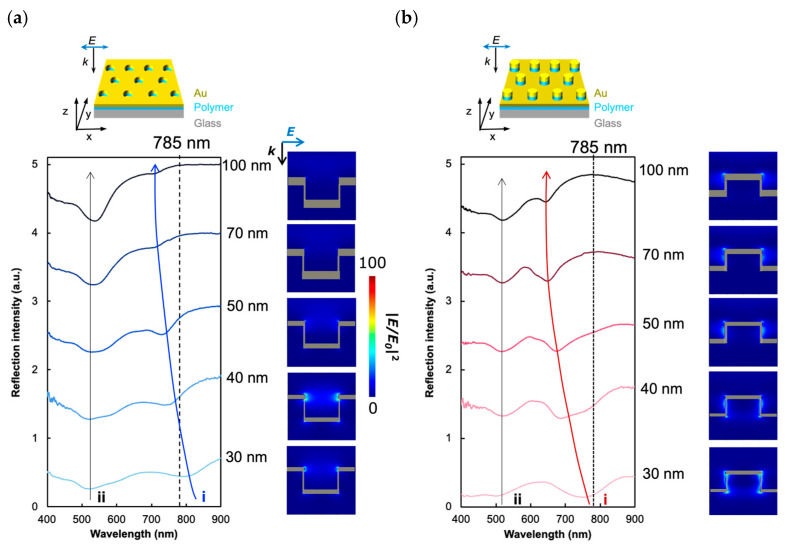
The optical characterization of PNHs and PNDs. The reflection spectrum of PNHs (**a**) and PNDs (**b**) with gold layer thicknesses ranging from 30 nm to 100 nm. The arrows indicate plasmonic modes i (blue for PNHs and red for PNDs) and ii (grey). The dashed line indicates the wavelength of 785 nm. The x-z cross-sectional enhanced electric field distributions expressed by |*E/E*_0_|^2^ at 785 nm are represented for each thickness of the gold layer.

**Figure 3 sensors-24-04599-f003:**
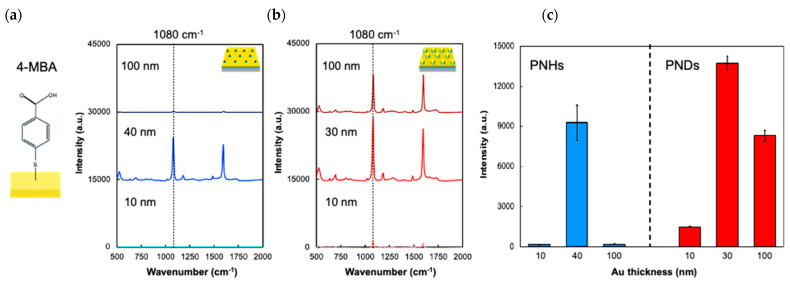
The comparison of the enhanced Raman intensity of 4−MBA by PNHs and PNDs with varied gold layer thickness. (**a**,**b**) The Raman spectrum with PNHs (10, 40, 100 nm) (**a**) and PNDs (10, 30, 100 nm) (**b**). The dashed line indicates 1080 cm^−1^, which is attributed to C−S (aromatic). (**c**) The comparison of Raman intensity at 1080 cm^−1^ between PNHs and PNDs. The Raman spectrum averaged 20 points with N.A. 0.8 objective lens. The error bars indicate standard deviation (SD).

**Figure 4 sensors-24-04599-f004:**
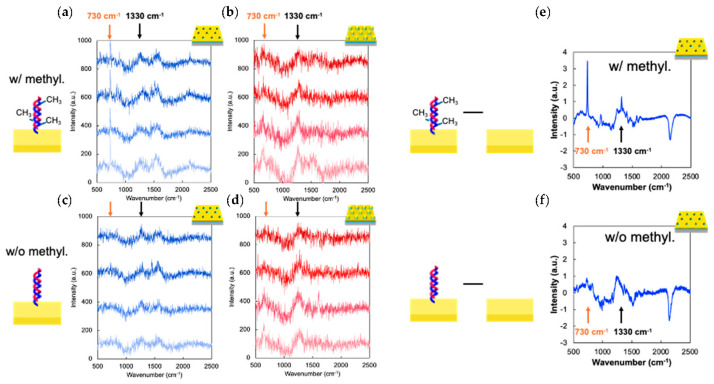
The identification of DNA methylation. (**a**,**b**) The Raman spectrum of methylated DNAs with PNHs (**a**) and PNDs (**b**) obtained from 4 separate samples. The orange and black arrows indicate 730 cm^−1^ and 1330 cm^−1^, respectively. (**c**,**d**) The Raman spectrum of unmethylated DNAs with PNHs (**c**) and PNDs (**d**). (**e**,**f**) The Raman spectrum obtained by subtracting that of bare PNHs from that of DNAs with PNHs.

## Data Availability

The data are contained within the article and Appendix A.

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
