# Peer review of "Nanoimprinted Plasmonic Crystals for Cost-Effective SERS Identification of Methylated DNAs"

_sensors, 2024, doi:10.3390/s24144599_

Round 1

Reviewer 1 Report

Comments and Suggestions for Authors

In this work, the authors have demonstrated rapid detection of methylated DNA in APC sequences  using nanoimprinted plasmonic crystals. The plasmonic resonance wavelengths of the PNHs and PNDs were adjusted to 785 nm by tuning the thicknesses of the Au layers. Evaluation of the SERS properties of PNHs and PNDs revealed that PNHs could enhance molecular Raman scattering by their plasmonically enhanced electric field, whereas PNDs  could enhance it by their plasmonic and photonic resonance modes. There are some points to be clarified and revised by the authors before acceptance.

(1)   Why is the Raman bands near 730 cm^-1 so narrow  in Figure 4a?

(2)   Figure 4, those intensities of Raman band are so weak, and background noise are so strong. How to clearly exhibit the identification of DNA methylation?

Author Response

Reviewer 1

In this work, the authors have demonstrated rapid detection of methylated DNA in APC sequences using nanoimprinted plasmonic crystals. The plasmonic resonance wavelengths of the PNHs and PNDs were adjusted to 785 nm by tuning the thicknesses of the Au layers. Evaluation of the SERS properties of PNHs and PNDs revealed that PNHs could enhance molecular Raman scattering by their plasmonically enhanced electric field, whereas PNDs could enhance it by their plasmonic and photonic resonance modes. There are some points to be clarified and revised by the authors before acceptance.

  • Why is the Raman bands near 730 cm^-1 so narrow in Figure 4a?

Thank you very much for your question. We could have not clarified that point. Therefore, we note possible reasons in following; We considered that the methylation attributed Raman peak intensity or band width are related to the DNA sequence used in experiments, and we expected that internal relaxation is suppressed by some chemical cause near 730 cm^-1. In this study, we have observed two typical Raman peaks which can be attributed to methylations according to some previous reports, however, their Raman shifts are varied a little in each report, and we could not observe another methylation attributed Raman peaks which were observed in another report. Therefore, we consider more detailed and massive experimental results will be needed to clarify these points.

  • Figure 4, those intensities of Raman band are so weak, and background noise are so strong. How to clearly exhibit the identification of DNA methylation?

Thank you for pointing this out. We agree the weakness of Raman band intensity and strongness of background noise. On the other hand, discussion about the capability to identify DNA methylation was based on results shown in Fig. 4b. In the Raman spectra in Fig 4b, the background noise was suppressed by extraction of bare sample from DNA sample, then obvious methylation attributed Raman peaks could be observed only in DNA methylation sample. Therefore, we consider that the identification of DNA methylation was clearly exhibited.

Reviewer 2 Report

Comments and Suggestions for Authors

The presented approach to SERS is highly promising. However, the analysis of the detection needs to be described more thoroughly.

Is it possible to fit the Raman spectra? Additionally, how were the peaks characterized? Have you considered using different wavelengths for Raman spectroscopy? If so, do you have any suggestions or recommendations regarding this?

Author Response

Reviewer 2

The presented approach to SERS is highly promising. However, the analysis of the detection needs to be described more thoroughly.

Is it possible to fit the Raman spectra? Additionally, how were the peaks characterized? Have you considered using different wavelengths for Raman spectroscopy? If so, do you have any suggestions or recommendations regarding this?

Thank you for your comments. We have answered to each question below.

(1) Is it possible to fit the Raman spectra?

In this study, we did not fit Raman spectra because the shape of spectra was clear. However, fitting Raman spectra with a Gaussian or Lorenz function may lead a less noisy spectrum. We think fitting process is more effective in case of overlap of spectral peaks.

(2) how were the peaks characterized?

We focused on the wavelength of the peak that changes with methylation, referring to a previous paper. Two peaks derived from the methylation were observed in this study. Therefore, we discussed them.

(3) Have you considered using different wavelengths for Raman spectroscopy?

Excitation wavelength is strongly limited by the measuring substrate. The excitation wavelength must be far enough from the plasma frequency of Au (-520 nm). Therefore, the wavelength of 532 nm cannot be used as the excitation wavelength for Au-based measurement substrates. Since photon intensity is weak at longer than 785 nm, the excitation wavelength options are 633 or 785 nm. In this study, the tunable wavelength was 785 nm in both the plasmonic nanodisks and plasmonic nanoholes.

Reviewer 3 Report

Comments and Suggestions for Authors

This study focuses on the preparation of plasmonic substrates based on nanoimprinting technology. The resulting substrates were used for direct identification of methylated DNA with adenomatous polyposis coli gene sequence. Before recommending this manuscript for publication in Sensors, the authors are encouraged to consider a number of comments below.

1. Line 22, remove “keyword 1”.

2. Lines 46-52. There are no references confirming the benefits of plasmonic arrayed systems.

3. Line 94. The development of a substrate where incubation for one day is required to measure the analyte seems to be an unpromising direction. How can authors explain such a different reaction time between the substrate and the analyte, namely 1 day with mercaptobenzoic acid and 1 hour with methylated DNA?

4. Line 102. The materials and methods do not contain data on the preparation of disulfide-modified DNA samples. In addition, from the data presented in Table S1, the probe is thiolated DNA. Please comment.

5. Line 125 Provide the full name for the abbreviation “SLRs mode”.

6. Line 131, 140 typos in the sentences, for example “indicatessan” or “SRES”. Correct here, as well as other typos in the text.

7. Lines 194-195 contain repeated sentences.

8. Section 3.2. The authors are recommended to calculate the enhancement factor for two obtained substrates with different thicknesses of the gold layer, which will allow them to evaluate their enhancing properties.

9. What is the rationale for choosing a DNA concentration of 1 nM? What concentrations are needed to identify methylated DNA with APC gene sequences?

10. Lines 211-225: In the second part of Section 3.3, which describes the results of amplification of the SERS signal for methylated and unmethylated DNA, there are insufficient references to the literature confirming the assumption of thermal diffusion processes. Please add them to the text of the manuscript.

11. The effect of gold layer thickness on the PNHs and PNDs on the enhancement of the Raman scattering of 4-MBA was investigated and 40 nm and 30 nm were chosen as an optimal, respectively. However, the complete disappearance of the SERS signal when 100 nm gold is deposited on PNHs is unclear. How can this be explained?

12. Figure 1. Please check the plasmonic crystal designations.

Author Response

Reviewer 3

This study focuses on the preparation of plasmonic substrates based on nanoimprinting technology. The resulting substrates were used for direct identification of methylated DNA with adenomatous polyposis coli gene sequence. Before recommending this manuscript for publication in Sensors, the authors are encouraged to consider a number of comments below.

(1) Line 22, remove “keyword 1”.

Thank you for pointing that out. We have removed it.

(2) Lines 46-52. There are no references confirming the benefits of plasmonic arrayed systems.

Thank you for pointing that out. We have added references.

(3) Line 94. The development of a substrate where incubation for one day is required to measure the analyte seems to be an unpromising direction. How can authors explain such a different reaction time between the substrate and the analyte, namely 1 day with mercaptobenzoic acid and 1 hour with methylated DNA?

Thank you for pointing that out. Mercaptobenzoic acid was used for evaluation of Raman enhancement by plasmonic crystals. We had no data about sufficient time to modify it on the gold surface, therefore, we had chosen excess time (overnight) to incubate. On the other hand, according to our previous experimental results, modification of probe DNA and hybridization of target DNA with probe DNA had been completed in one hour.

(4) Line 102. The materials and methods do not contain data on the preparation of disulfide-modified DNA samples. In addition, from the data presented in Table S1, the probe is thiolated DNA. Please comment.

Thank you for pointing that out. Thiolated DNA is correct. We have revised “disulfide” to “thiolated” (See line 102.). All DNA sample was purchased from BEX Co. Ltd. (Toyota, Japan). We have added the explanation (See line 106).

(5) Line 125 Provide the full name for the abbreviation “SLRs mode”.

Thank you for pointing that out. We have added full name for that (See line 125.).

(6) Line 131, 140 typos in the sentences, for example “indicatessan” or “SRES”. Correct here, as well as other typos in the text.

Thank you for pointing that out. We have corrected them (See line 131, 140.).

(7) Lines 194-195 contain repeated sentences.

Thank you for pointing that out. We have corrected that point (See line 194.).

(8) Section 3.2. The authors are recommended to calculate the enhancement factor for two obtained substrates with different thicknesses of the gold layer, which will allow them to evaluate their enhancing properties.

Thank you for recommendation. We do agree that point. Generally, Raman enhancement factors by plasmonic substrates are based on comparison between nanostructured and nonstructured surface. Actually, we tried to calculate and compare the enhancement factors between mercaptobenzoic acids and DNAs, however, Raman bands of DNA samples were difficult to observe with flat gold layer substrate. On the other hand, as discussed in the paper, enhancement factors of mercaptobenzoic acids did not reflect them of DNA samples, therefore, we consider that the evaluation of enhancement factors only by mercaptobenzoic acids are not enough to generally discuss about enhancement factors for DNA observation. Therefore, we have not mentioned about that point in the manuscript.

(9) What is the rationale for choosing a DNA concentration of 1 nM? What concentrations are needed to identify methylated DNA with APC gene sequences?

Thank you for question. First, we should correct a mistake in description of DNA concentration. We have corrected that from 1 nM to 1 mM (See line 101, 104). Under estimation of standard DNA concentration as 10 – 100 ng/ml corresponding to 0.3 – 3 nM, 1 nM is a reasonable concentration. In this experiment, we used enough high concentrated DNA sample in order to reliably evaluate SERS properties of plasmonic crystals. We have already confirmed that 1 nM APC gene can be detected. We plan to evaluate the limit of detection and quantitativity of APC gene.

(10) Lines 211-225: In the second part of Section 3.3, which describes the results of amplification of the SERS signal for methylated and unmethylated DNA, there are insufficient references to the literature confirming the assumption of thermal diffusion processes. Please add them to the text of the manuscript.

Thank you for advice. We estimate your point is the thermal diffusion of plasmonic substrate. As far as we know, there are no reports about comparison of thermal diffusion and its effect on SERS properties between open system (nanoholes) and closed system (nanoparticles). It is assumed that plasmonically activated electrons in “gold plate” can be diffused faster than those in “gold particles”, which partially determine the SERS properties. Our results are considered to be suggestion about this point. We have corrected explanation about this point. If you have any insights into this discussion, we would be grateful if you could provide them.

(11) The effect of gold layer thickness on the PNHs and PNDs on the enhancement of the Raman scattering of 4-MBA was investigated and 40 nm and 30 nm were chosen as an optimal, respectively. However, the complete disappearance of the SERS signal when 100 nm gold is deposited on PNHs is unclear. How can this be explained?

Thank you for question. The SERS signal by PNHs with 100 nm thick gold layer could be slightly observed (See Fig. 3c). The weakness of signal can be attributed to the low electric field enhancement by PNHs with 100 nm thick gold layer (See Figure 2a, electric field distribution of PNHs with 100 nm thick gold layer.)

(12) Figure 1. Please check the plasmonic crystal designations.

Thank you for pointing that out. We have corrected that. See Figure 1.

Reviewer 4 Report

Comments and Suggestions for Authors

The manuscript by Endo et. al developed a plasmonic SERS substrate via nanoimprinting technique for DNA methylation detection. There are some questions to authors that I suggest to address for further improvement.

1.      The author should more specifically summarize recent researches about the nanoimprinting technique used in fabrication of SERS substrate in the Introduction. Plasmonic nanoholes and nanodisks have been developed for many years.

2.      I don’t see why nanoimprinting technique is cost-effective for plasmonic substrate. Do you check the cost? To my knowledge, Au layer deposition is a high-cost procedure.

3.      Since the idea of this work is to develop a plasmonic SERS substrate for DNA methylation assay, the enhancement factor of the substrate should be calculated.

4.      What are the periodic parameters of the plasmonic array? How does the arrangement of nanoholes or nanodisks affect the plasmonic property?

5.      Can you provide the details about which bond or vibrations 730 and 1330 cm-1 belong to in Figure 4? Can you give a reason why the peak at 730 cm-1 is more observable on the PNH substrate than the PND?

6.      Why the PNH substrate is “open” plasmonic system while the PND is “closed”? Please explain.

7.      Does the concentration of methylated DNA influence the SERS signal?

8.      In Figure 1 there are some typos. Labels of the two substrate sketches are the same.  

Author Response

Reviewer 4

The manuscript by Endo et. al developed a plasmonic SERS substrate via nanoimprinting technique for DNA methylation detection. There are some questions to authors that I suggest to address for further improvement.

(1) The author should more specifically summarize recent researches about the nanoimprinting technique used in fabrication of SERS substrate in the Introduction. Plasmonic nanoholes and nanodisks have been developed for many years.

Thank you for pointing out that. We have added a sentence and related references. See line 55.

(2) I don’t see why nanoimprinting technique is cost-effective for plasmonic substrate. Do you check the cost? To my knowledge, Au layer deposition is a high-cost procedure.

Thank you for pointing that out. Our imprinted plasmonic crystals can be massively produced by using only one mold which can be reusable. Thus, the cost for producing a lot of plasmonic crystals can be reduced in term of industrial application.

(3) Since the idea of this work is to develop a plasmonic SERS substrate for DNA methylation assay, the enhancement factor of the substrate should be calculated.

Thank you for recommendation. We do agree that point. Generally, Raman enhancement factors by plasmonic substrates are based on comparison between nanostructured and nonstructured surface. Actually, we tried to calculate and compare the enhancement factors between mercaptobenzoic acids and DNAs, however, Raman bands of DNA samples were difficult to observe with flat gold layer substrate. On the other hand, as discussed in the paper, enhancement factors of mercaptobenzoic acids did not reflect them of DNA samples, therefore, we consider that the evaluation of enhancement factors only by mercaptobenzoic acids are not enough to generally discuss about enhancement factors for DNA observation. Therefore, we have not mentioned about that point in the manuscript.

(4) What are the periodic parameters of the plasmonic array? How does the arrangement of nanoholes or nanodisks affect the plasmonic property?

Thank you for question. The pitch was approximately 440 nm, see line 113. The pitch of the crystal affects the plasmonic resonance wavelength. In this study, we used one mold for fabricating plasmonic crystals thus the pitch was fixed.

(5) Can you provide the details about which bond or vibrations 730 and 1330 cm-1 belong to in Figure 4? Can you give a reason why the peak at 730 cm-1 is more observable on the PNH substrate than the PND?

Thank you for pointing that out. First, we response to the former issue. According to the report; https://doi.org/10.1016/j.talanta.2022.123941, it is considered that the Raman peak at 730 cm-1 and 1330 cm-1 can be attributed to ring breathing vibration of adenine and guanine, respectively. They mentioned that the ring breathing vibration of adenine at 730 cm-1 is affected by methylation of cytosine and showed that the Raman peak intensity at 730 cm-1 of cytosine methylated DNA was higher than that of unmethylated DNA. However, Raman peak wavenumber which would be attributed to cytosine methylation are varied in many reports, and some reports did not attribute the wavenumber to any mode. At least, our study demonstrated that DNA methylation could be identified based on the observation of Raman peak at 730 cm-1. We have added the explanation about this in the manuscript, see line 196.

Next, we answer to the latter question. At this stage, we again cannot give a clear reason for your question. This result was unexpected from the evaluation of SERS properties with 4-MBA because the PNDs offered higher enhancement of Raman than the PNHs. The consideration for your question is described in the manuscript, see line 218-225.

(6) Why the PNH substrate is “open” plasmonic system while the PND is “closed”? Please explain.

Thank you for question. Generally, the plasmonic resonance in PNHs are based on both localized and propagating mode while that in PNDs are based on only localized mode. Therefore, we can call the former as “open” and the latter as “closed”.

(7) Does the concentration of methylated DNA influence the SERS signal?

Thank you for question, and exactly right. In the next step, we should investigate the SERS-based quantitativity to methylated DNA by utilizing our plasmonic crystals.

(8) In Figure 1 there are some typos. Labels of the two substrate sketches are the same.  

Thank you for pointing that out. We have modified that.

Round 2

Reviewer 3 Report

Comments and Suggestions for Authors

The authors took into account the comments, so I recommend this manuscript for publication in the Sensors journal.